# Investigating Combined Drought- and Heat Stress Effects in Wheat under Controlled Conditions by Dynamic Image-Based Phenotyping

**Lamis Osama Anwar Abdelhakim** [1,*] **, Eva Rosenqvist** [2], **Bernd Wollenweber** [3] **, Ioannis Spyroglou** [4] **, Carl-Otto Ottosen** [1] **and Klára Panzarová** [5]

1 Department of Food Science- Plant, Food & Climate, Aarhus University, Agro Food Park 48, DK-8200 Aarhus N, Denmark; coo@food.au.dk
2 Department of Plant and Environmental Sciences- Crop Sciences, University of Copenhagen, Højbakkegård Allé 13, DK-2630 Tåstrup, Denmark; ero@plen.ku.dk
3 Department of Agroecology- Crop Health, Aarhus University, Forsøgsvej 1, DK-4200 Slagelse, Denmark; bernd.wollenweber@agro.au.dk
4 Plant Sciences Core Facility, Central European Institute of Technology, Masaryk University, Kamenice 5, 62500 Brno, Czech Republic; ioannis.spyroglou@ceitec.muni.cz
5 Photon Systems Instruments, (PSI, spol. sr. o.), 66424 Drásov, Czech Republic; panzarova@psi.cz
* Correspondence: lamis@food.au.dk

**Abstract:** As drought and heat stress are major challenges for crop productivity under future climate changes, tolerant cultivars are highly in demand. This study investigated the potential of existing Nordic wheat genotypes to resist unfavorable conditions. Four genotypes were selected based on their heat sensitivity (heat-sensitive: LM19, SF1; heat-tolerant: LM62, NS3). At the tillering stage, the plants were subjected to four treatments under controlled conditions: control, drought, heat and combined drought and heat stress. The morpho-physiological performance was quantified during the early and late phase of stress, as well as the recovery phase. We applied an integrative image-based phenotyping approach monitoring plant growth dynamics by structural Red Green Blue (RGB) imaging, photosynthetic performance by chlorophyll fluorescence imaging and transpiration efficiency by thermal infrared imaging. The results demonstrated that the selected genotypes were moderately affected in their photosynthetic efficiency and growth under drought stress, whereas heat and combined stress caused rapid reductions in photosynthesis and growth. Furthermore, drought stress had a major impact on canopy temperature. The NS3 genotype was the most robust genotype, as indicated by its improved response under all stress treatments due to its relatively small biomass. However, the genotypes showed different tolerance to individual and combined stress.

**Keywords:** climate change; drought; heat stress; image-based phenotyping; RGB imaging; chlorophyll fluorescence imaging; thermal infrared imaging

## 1. Introduction

The increase in climatic variability, such as heat waves together with prolonged drought episodes, is a threat to agricultural productivity [1,2]. Sustaining the yield of economical crops such as wheat (*Triticum aestivum*) is crucial for the maintenance of food security [3]. Many studies reported that high temperatures above the optimum negatively affect both the yield and quality of crops [4,5]. Therefore, robust wheat varieties that can acclimate to adverse climate conditions are becoming a priority for crop breeding and production [6]. The effect of abiotic stress is dependent on its intensity, frequency and duration, as well as on the genotype used and the developmental stage of the crop [7]. As plants have developed several adaptive physiological and metabolic mechanisms, the sensitivity of genotypes to stress episodes varies [8].

Drought is one of the limiting factors on plant growth due to the reduction of primary metabolic processes and the need to enhance the synthesis of metabolites for osmotic adjustment [9]. During mild to moderate drought stress, stomatal closure is the earliest response, which reduces intercellular $CO_2$ levels but not the photosynthetic rate, and with increasing drought severity, the photosynthetic rate is reduced [10,11]. In addition, plants adapt to drought stress by reducing their leaf area and by decreasing both stomatal conductance and transpiration rates to reduce water loss [12]. Under drought stress conditions, plants regulate the amount of photosynthetic light-harvesting pigments by lowering the photosynthetic electron transport by adjusting the chlorophyll antenna size [13].

High temperatures result in changes in morphological, physiological and biochemical processes [14]. Under heat stress, plants have developed mechanisms of tolerance via short-term mechanisms of avoidance or acclimation and/or long-term mechanisms of phenological and morphological adaptations [15]. Thermotolerance is one of the heat-resistant mechanisms induced in response to heat stress to protect the photosynthetic apparatus [16,17]. Moderate heat stress causes slight or no damage to photosystem II, despite reductions in the photosynthetic capacity [18]. Wheat has the ability to maintain a high activation state of Rubisco as a high-temperature acclimation mechanism [19]. In addition, plants can avoid damage by heat stress through increasing both stomatal conductance and transpiration rates, thereby enhancing leaf cooling [20]. However, the variation among wheat genotypes in terms of their photosynthetic thermal acclimation under high temperature is still unclear [6].

Intensive studies have been investigating the physiological response of crops to individual stress types, such as heat [15,21] or drought [22,23]. However, crops respond differently to combined abiotic stress types [24,25]. Concurrently occurring abiotic stress events result in complex responses and interacting signaling pathways that may inhibit each other [26]. Under heat stress, plants with adequate water supply keep their stomata open to enhance leaf cooling through transpiration, while under water scarcity plants close their stomata to avoid excess water-loss resulting in increased leaf temperature [16]. In tobacco and wheat, combined drought and heat stress has more severe effects on the rates of growth and development, as well as physiological traits such as photosynthetic rate, stomatal conductance and chlorophyll fluorescence, in comparison to the effect of individual stress types [27,28]. Under combined stress, the plants' response is affected by the most intense stress type [29].

Plant phenomics is the study of plant composition, development and performance by using phenotyping systems to screen desired agricultural traits under unfavorable conditions [30]. It is an integrative method for the dynamic assessment of the stress response with possibilities for high time-resolution. The use of integrative high-throughput plant phenotyping platforms with imaging sensors, robotics and computer-vision facilitates dynamic, non-destructive measurements of various plant traits [31–33]. Evaluating plants' performances under abiotic stress by screening phenotypic traits such as growth rate, photosynthetic activity and temperature tolerance is essential to identify stress-tolerant cultivars [34]. Plant shoot biomass, growth rate and development over time can be quantified by using digital color images via Red Green Blue (RGB) imaging [32]. Moreover, digital imaging can visualize changes in leaf color to detect leaf senescence [30]. Chlorophyll fluorescence as a non-invasive measurement of photosystem II (PSII) activity is used to monitor the performance of the plant photosynthetic apparatus and to detect the damage to PSII under stress conditions [34,35]. The monitoring of the maximum quantum efficiency of PSII photochemistry ($F_v/F_m$) and of PSII operating efficiency ($F_q'/F_m'$) is useful for the rapid screening of tolerance to abiotic stress [36]. The difference between air and leaf temperature is known as canopy temperature depression, and reflects the tolerance under drought and heat stress [37]. Using infrared thermography, canopy temperature is measured to reflect the stomatal regulations and leaf cooling capacity [37,38].

More recent studies, under both field conditions [39–42] and controlled conditions, where an individual abiotic stress episode is applied such as drought [42,43] or heat [44],

have used high-throughput phenotyping to screen for desirable traits of climate-tolerant wheat varieties. However, our understanding of the underlying mechanisms of the plants' response under combined drought and heat stress in contrasting wheat genotypes by an image-based phenotyping approach is still limited, and it is complicated to perform in the field. Our hypothesis has been that heat-sensitive and heat-tolerant genotypes of wheat will reflect different susceptibilities to individual and combined stress types. The aim of this study was to understand the underlying response mechanisms occurring during early and late phases of abiotic stress in contrasting wheat genotypes, by monitoring morpho-physiological traits using dynamic image-based phenotyping under controlled conditions.

## 2. Materials and Methods

### 2.1. Plant Material and Growing Conditions

Four Nordic wheat (*Triticum aestivum*) genotypes were selected in this study, based on their $F_v/F_m$ from a previous screen experiment where plants were induced to heat stress at 40 °C for four days (unpublished data). The selected heat-sensitive genotypes were LM19 (breeding line, Lantmännen, Svalöv, Sweden) and SF1 (Trappe, Sejet Plant Breeding, Horsens, Denmark), and the heat-tolerant genotypes were LM62 (breeding line, Lantmännen, Svalöv, Sweden) and NS3 (Skagen, Nordic Seed, Dyngby, Denmark). Two seeds were sown in 3.1 L pots (15.5 cm diameter, 20.5 cm height) filled with 1000 g commercial peat substrate (Pindstrup Færdigblanding 2, Pindstrup Mosebrug A/S, Ryomgaard, Denmark) and watered to full soil water holding capacity (ca. 2000 g). The plants were grown in a climate-controlled growth chamber (FS_WI, Photon Systems Instruments (PSI), Drásov, Czech Republic) under long day conditions (16 h photoperiod). The climate conditions in the growth chamber were set at 22/19 °C for day/night temperature with 60% relative humidity (RH) and 350 µmol m$^{-2}$ s$^{-1}$ photosynthetic photon flux density (PPFD) (336 µmol m$^{-2}$ s$^{-1}$ cool-white LED and 14 µmol m$^{-2}$ s$^{-1}$ far-red LED lighting as determined with SpectraPen MINI (PSI, Drásov, Czech Republic)).

Seeds were sown in two batches (38 pots per batch per genotype) in order to be able to measure the plants at the same developmental stage, and two treatments were applied per batch. At 13 days after sowing (Zadoks 12) [45], one seedling was kept per pot by thinning and a blue rubber mat was added per pot to cover the soil surface. At 14 days after sowing, plants were fertigated by flooding for 20 min with a commercial nutrient solution mix (Kristalon Podzim, AGRO CS, Říkov, Czech Republic) once a week and then 10 min before starting the treatments.

### 2.2. High-Throughput Phenotyping

In order to characterize the morphological and physiological performance of the plants, a parallel monitoring system of plant growth dynamics (by structural RGB imaging), photosynthetic performance (by chlorophyll fluorescence imaging) and transpiration efficiency (by thermal infrared imaging) was set up, with all sensors for digital analysis being implemented in the PlantScreen$^{TM}$ Modular system (PSI, Drásov, Czech Republic). The plants were transferred from the growth chamber to the PlantScreen$^{TM}$ Modular system for the phenotyping measurements. Each pot was loaded onto a transport disk automatically moving on a conveyor belt between the acclimation unit, robotic-assisted imaging units and the weighing and watering unit (Figure 1A).

### 2.3. Treatments

At 27 days after sowing (Zadoks 23–25), the control and drought stress treatments were applied in the first batch. The heat and combined (drought + heat) stress treatments at 36/26 °C day/night temperature with 40/60% day/night RH were applied in the second batch. Two genotypes (LM19 and NS3) grown in the second batch unintentionally had a PPFD of 150 µmol m$^{-2}$ s$^{-1}$ until 3 days before the measurements, when the PPDF was adjusted to 300 µmol m$^{-2}$ s$^{-1}$. The irrigation protocol was controlled by using the automated watering and weighing unit of the PlantScreen$^{TM}$ Modular system (PSI,

Drásov, Czech Republic). The pots were watered to the reference weight where the soil relative water content (SRWC) was maintained at 70% for control and heat treatments while withholding irrigation for drought and combined treatments until SRWC reached 20–30% [46]. The water consumption was calculated based on the pot weight before and after irrigation between two consecutive days. The drought stress was induced for 16 days while the heat and combined stress were induced for 11 days followed by 3 days of recovery (Figure 1B).

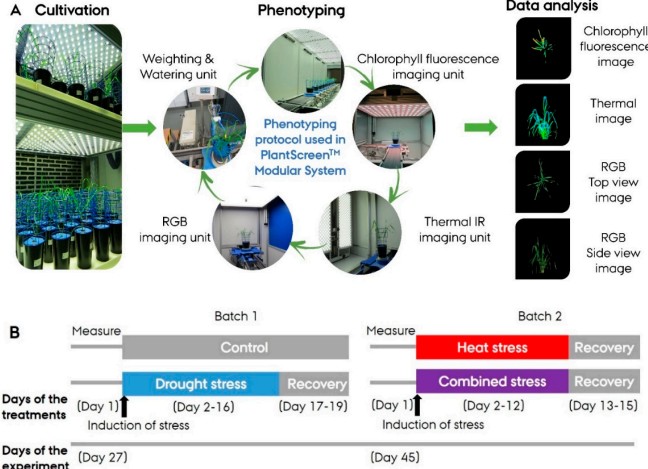

**Figure 1.** (**A**) Scheme shows the screening process started by manually adding the plants from the growing chamber into the PlantScreen^TM Modular system, the applied phenotyping protocol and the output of analysis raw images followed by exporting the data. (**B**) Timeline of applying the four treatments where two treatments were applied per batch including day 1 before inducing the stress and days after inducing the stress treatments, followed by three days of recovery and days of the whole experiment for both batches.

In the first batch, blue plant stick supports were added to hold the plant vertically before measurements, but at day 16, the sticks were replaced with plant grid supports. In the second batch, blue plant grid supports were added before measurements started to avoid any mechanical damage during the automated movement in the PlantScreen^TM Modular system (PSI, Drásov, Czech Republic).

*2.4. Phenotyping Protocol*

The measurements started with the light or dark adaptation of the plants in the adaptation tunnel, as specified further. In the morning at 8:00, light adapted plants were measured on four rounds, each with 10 plants per round. The rounds were randomized along the days of phenotyping. After adaptation, measurements started by chlorophyll fluorescence imaging, IR imaging, RGB imaging, and ended by weighting and watering the individual plants. A total time of 45 min was required to measure 10 plants. In the afternoon at 12:00, plants were dark-adapted for 25 min in the adaptation tunnel, then chlorophyll fluorescence and light response curves were measured by chlorophyll fluorescence imaging followed by weighting and watering. A total time of 65 min was required to measure 10 plants. Each treatment had 5 replicates per genotype in the morning measurements and 4 replicates per genotype in the afternoon measurements. Plants were monitored three times a week under control and drought stress and on a daily basis under heat and combined stress treatments. The raw data were automatically processed through the PlantScreen^TM Analyzer software (PSI, Drásov, Czech Republic).

2.4.1. Chlorophyll Fluorescence Imaging

Non-invasive assessment of photosynthetic activity was measured by using an enhanced version of the FluorCam FC-800MF pulse amplitude modulated (PAM) chlorophyll

fluorometer (PSI, Drásov, Czech Republic). In the chlorophyll fluorescence imaging unit, a high-speed charge-coupled device (CCD) camera (resolution of $720 \times 560$ px, frame rate 50 fps and 12-bit depth) is mounted on a robotic arm positioned in the middle of the LED light panel. The LED panel was equipped with $3 \times 64$ orange-red (618 nm) and 64 cool-white LEDs (6500 K), distributed equally over $75 \times 75$ cm [47]. Measurements on light- and dark-adapted plants were conducted for all treatments.

Light-adapted measurements:

At the adaptation tunnel, plants were adapted for 5 min at 300 μmol m$^{-2}$ s$^{-1}$ (because plants were transferred manually from the climate chamber to the PlantScreen$^{TM}$ modular system) followed by 3 min at 750 μmol m$^{-2}$ s$^{-1}$ cool-white actinic light, then each plant was automatically transported one by one to the light-isolated chlorophyll fluorescence imaging unit for measurement. The measuring protocol was set as the following: 3 s of cool-white actinic light at 750 μmol m$^{-2}$ s$^{-1}$ was applied to determine the steady state F-level in the light-adapted state ($F_t'$), followed by an 800 ms saturation pulse of 3350 μmol m$^{-2}$ s$^{-1}$ to determine the maximum fluorescence in the light-adapted state ($F_m'$). Afterwards, the actinic light was turned off, far-red (735 nm) turned on and PSII was relaxed in the dark for 800 ms to determine the minimum fluorescence in the light-adapted state ($F_o'$). The PSII operating efficiency ($F_q'/F_m'$) and the fraction of open PSII centers (oxidized $Q_A$) ($q_L$) were measured at a PPFD of 750 μmol m$^{-2}$ s$^{-1}$ and calculated according to Baker [35].

Dark-adapted measurements and light response curves:

In the afternoon measurements, plants were dark-adapted for 25 min at the adaptation tunnel and then each plant was automatically transported to the light-isolated chlorophyll fluorescence imaging unit. The maximum quantum efficiency of the PSII photochemistry ($F_v/F_m$) and the light response curve were measured as the following: 5 s non-actinic measuring light was applied to measure the minimum fluorescence in the dark-adapted state ($F_o$) followed by an 800 ms saturation pulse of 3350 μmol m$^{-2}$ s$^{-1}$ to determine the maximum fluorescence in the dark-adapted state ($F_m$). Afterwards, four actinic light levels were applied in the light response curve protocol (Light steady state (Lss) 1, 300 μmol m$^{-2}$ s$^{-1}$; Lss 2, 500 μmol m$^{-2}$ s$^{-1}$; Lss 3, 750 μmol m$^{-2}$ s$^{-1}$; and Lss 4, 1000 μmol m$^{-2}$ s$^{-1}$). A duration of 120 s for Lss1 and 60 s for the other light level allowed the plants to reach steady state. The selected durations were based on previous light response curves measured on dark-adapted leaves using a Mini-PAM (Walz GmbH, Effeltrich, Germany) to establish the duration required to reach steady state for each light level (data not shown).

### 2.4.2. Visible RGB Imaging

RGB imaging was used to assess digital biomass based on side view imaging from two angles (0° and 90°) (RGB1) and top view imaging (RGB2). In the RGB imaging unit, two RGB cameras (top and side) were mounted on robotic arms, a rotating table with precise angle positioning, and LED-based lighting source supplemented with each camera to ensure homogeneous illumination of the imaged plant. RGB images (resolution $2560 \times 1920$ pixels) of the plants were captured using the GigE uEye UI-5580SE-C-5 Megapixels QSXGA camera with 1/200 CMOS sensor (IDS, Germany) from side and top view. For the side view projections, line scan mode was used with a resolution of 2560 px/line, 200 lines per second [48]. The imaged area from the side view was $1205 \times 1005$ mm (height $\times$ width), while the imaged area from the top view position was $800 \times 800$ mm [48]. The digital biomass of each plant was automatically extracted with plant data analyzer software (PSI, Drásov, Czech Republic) and processed as described by Awlia et al. [47]. The *plant volume* [49] and *relative growth rate* [48] were calculated as:

$$Plant\ volume = \sqrt{Area_{Side\ view}^2 \times Area_{Top\ view}} \tag{1}$$

$$Relative\ growth\ rate = \frac{ln\ (Volume)_{T_2} - ln\ (Volume)_{T_1}}{T_2 - T_1} \tag{2}$$

where $T_1$ and $T_2$ indicate the time interval (days).

### 2.4.3. Thermal Imaging

Thermal infrared imaging was used to assess canopy temperature based on the side view imaging. The thermal imaging unit included a side view thermal camera mounted on a robotic arm, a rotating table with precise plant positioning and a background wall with an integrated temperature sensor to increase the contrast during the image processing. The thermal images were acquired in darkness using line scan mode with a scanning speed of 50 Hz and each line consisting of 710 pixels [50]. The imaged area was $1205 \times 1005$ mm (height $\times$ width). The canopy temperature of each plant was automatically extracted with plant data analyzer software (PSI, Drásov, Czech Republic) by mask application, subtraction of background, and pixel-by-pixel integration of values across the whole plant surface area (Figure S1A,B). The absolute canopy temperature of each plant ($T_{canopy}$) was normalized with the air temperature inside the thermal imaging unit ($T_{air}$), known as $\Delta T$ (°C) [51], to reduce the effect of the difference in the image acquisition timing and temperature variability among individual plants.

### 2.5. Water Use Efficiency

The water use efficiency (*WUE*) was defined as the ratio of biomass accumulation (volume) to the *cumulative water amount used* and was given by the following formula [52]:

$$WUE = \frac{Biomass\left(Plant\ Volume\left(dm^3\right)\right)}{Cumulative\ water\ amount\ used\ every\ day}$$

### 2.6. Manual Measurements

The spectral reflectance of leaves was measured between 10:00 and 11:00 at the last day after inducing the stress treatments using a hand-held PolyPen RP 410 (PSI, Drásov, Czech Republic). Three measurements were taken on a young fully expanded leaf of the main tiller per plant. Nine plants per genotype per treatment were measured.

Plants were dark-adapted for 25 min in the afternoon and a hand-held PAM fluorometer FluorPen FP100 (PSI, Drásov, Czech Republic) was used to measure $F_v/F_m$. Three measurements were taken on a young fully expanded leaf of the main tiller per plant at the last day after inducing the stress treatments. Four plants per genotype per treatment were measured.

### 2.7. Leaf Pigment Analysis

One young fully expanded leaf of the main tiller was harvested and immediately frozen in liquid nitrogen after harvest at the last day after inducing the stress treatments. The harvested samples were lyophilized at $-96$ °C under pressure 1 hPa over 24 h and stored in a deep freezer at $-80$ °C. The samples were then subjected to chemical extraction of the plant pigments. The dry plant biomass (about 5 mg) was homogenized and repeatedly extracted with 0.5–1 mL of cooled methanol/acetone until the pellet turned white. Individual purified aliquots of extracts were pooled into the final sample. The sample was then immediately analyzed by a high-performance liquid chromatography method with diode array detection (HPLC/DAD) according to Garcia-Plazaola and Becerril [53]. HPLC analysis was performed using two mobile phases: (A) acetonitrile:methanol:0.1M TRIS (pH 8) = 42:1:7 and (B) methanol:ethyl acetate = 17:8. The extracted pigments in the sample were separated by gradient elution with a mobile phase flow rate of 1.2 mL/min on a Kinetex RP C18 chromatographic column, tempered at 25 °C. Qualitative and quantitative pigment results were obtained from chromatograms of samples measured using a DAD detector at 436 nm and 665 nm.

### 2.8. Data Analysis

The parameters of plants under control, drought, heat and combined stress treatments were analyzed by one-way analysis of variance (ANOVA) (Tukey's post hoc test) using SPSS 16.0 (SPSS Inc. Chicago, IL, USA). The parameters used in this study are given in

Table S1. The categorization of these parameters is given in a pie chart (Figure S2). The data contain a number of geometric, fluorescence, and color-, water- and temperature-related traits (Table S1). The correlation between all these traits was examined through correlation heatmap (Figure S3). Moreover, for the examination of differences between the means of genotypes for all the parameters that were presented in the figures in different time points (days after inducing stress), mixed models [54,55] and pairwise comparisons with the use of Tukey test were applied in R studio (R GUI 4.0.3) using the "lmer" and "emmeans" packages [56,57]. As variables, days after inducing stress, genotype and their interaction were included (Tables S2–S7). For Figure S7, light levels were used as variables instead of days after inducing stress (Table S8). Random (subject-specific) effects were used so the post-hoc pairwise comparisons test considered the variability that exists (means and errors) for each genotype in different time points. For Figure S7, random intercept was used as only one day was shown, and as a result the significance should be studied on a larger scale as homogeneity of errors-variances is assumed.

Additionally, growth modeling was performed. Two kinds of models that describe in a satisfying way the growth of the plants based on volume were used. These models are commonly applied for plant growth modeling [58]. The first model is a simple exponential growth model that is applied mostly for control treatments as without stress the plants grow exponentially. This model has the form of the following equation:

$$y = C\, e^{rx},$$

where $C$ is the starting point, $r$ is the growth rate and x denotes the day.

The previous equation can become the following:

$$\log(y) = \log(Ce^{rx})$$

$$\log(y) = \log C + \log(e^{rx})$$

$$\log(y) = \log C + \mathbf{r}x$$

The second model with best fit mostly for stress treatments is a bell-shaped model of the form:

$$\log(y) = logC + \mathbf{r}x + r_2 x^2$$

This model is used for the stress treatments as it also contains the coefficient $r_2$. During stress periods, $r_2$ is usually negative and shows how fast the growth curve starts to decay because of the induced stress. The coefficient of determination $R^2$ (highest) and Akaike information criterion (lowest) were used for the selection of the best growth model [59]. To have a clear image of which genotype is more resistant to stress, we scaled all the series to the first day before inducing stress, so that every series starts from the same point and the results are unbiased by the sizes of the plants. This allowed us to recognize which genotype was more resistant by localizing the growth curve that is above the others. The same approach was used for the growth modeling of *WUE*. Both exponential and bell-shaped curve models were used for *WUE*.

## 3. Results

### 3.1. Soil Relative Water Content

In the control treatment, the soil relative water content (SRWC) was maintained above 50% throughout the experiment (Figure 2A$_1$). With drought stress, the SRWC decreased significantly in LM19 and LM62 compared to SF1 and NS3 (Figure 2A$_2$). During heat and combined stress treatments, SRWC was the lowest in LM62 followed by LM19 and SF1, while NS3 showed significantly higher SRWC compared to the other genotypes (Figure 2A$_{3,4}$). The RGB side view images clearly showed the genotype-specific morphological differences already at day 1 before inducing the stress. Treatment-specific impacts on growth performances of the plants were observed in all genotypes across the early (first week) and late (second week) stress phases, followed by the recovery phase (Figure 2B$_{1-4}$ and Table S2).

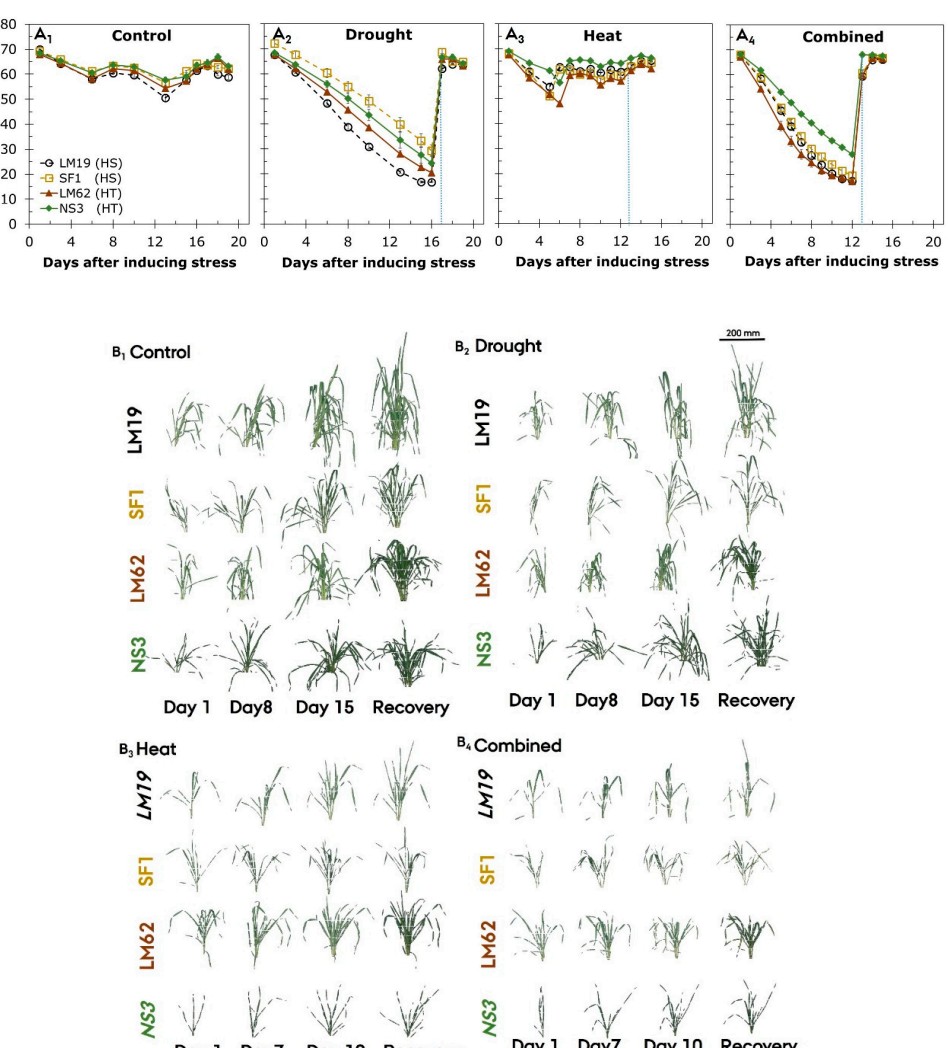

**Figure 2.** (**A₁₋₄**) Soil relative water content (SRWC) at day 1 before inducing the stress and along days after inducing the stress treatments followed by three days of recovery in four wheat genotypes. HS represents heat-sensitive genotypes (open symbols, dashed lines) and HT represents heat-tolerant genotypes (closed symbols, solid lines). Blue dotted line denotes the recovery phase. The data represent mean values ± standard error of mean (S.E.M.) (*n* = 5). (**B₁₋₄**) RGB images at control, drought, heat and combined stress treatments in four wheat genotypes during day 1, 8, 15 of control and drought stress and day 1, 7, 10 of heat and combined stress and third day of recovery. The plant batches for heat and combined stress in LM19 and NS3 (marked by italic letters) were unintentionally given half the growth light up to three days before the start of the stress treatment, which explains their smaller plant size at day 1. The scale bar refers to all RGB images in the figure.

*3.2. RGB Imaging*

The assessment of plant growth rate via RGB imaging was used to characterize differences between the genotypes (Figure 3A₁₋₄ and Table S3.1). In the control treatment, the relative growth rate (RGR) was similar in all genotypes (Figure 3A₁). With drought stress, RGR was significantly lower during the late stress phase only in heat-sensitive genotype LM19 compared to SF1. At the third day of recovery, RGR was significantly higher in NS3 compared to the other genotypes LM19 and SF1 (Figure 3A₂). With heat stress, RGR was significantly higher in the heat-tolerant genotypes LM62 and NS3 at the last day of stress (day 12) and third day of recovery, compared to the heat-sensitive genotypes LM19 and SF1 (Figure 3A₃). With combined stress, the RGR gradually decreased in all genotypes and at the last day of stress the RGR was significantly lower in LM62 and

SF1 compared to NS3 (Figure 3A$_4$). NS3 showed significantly higher RGR at the third day of the recovery phase compared to the other genotypes in all treatments (Figure 3A$_{1–4}$).

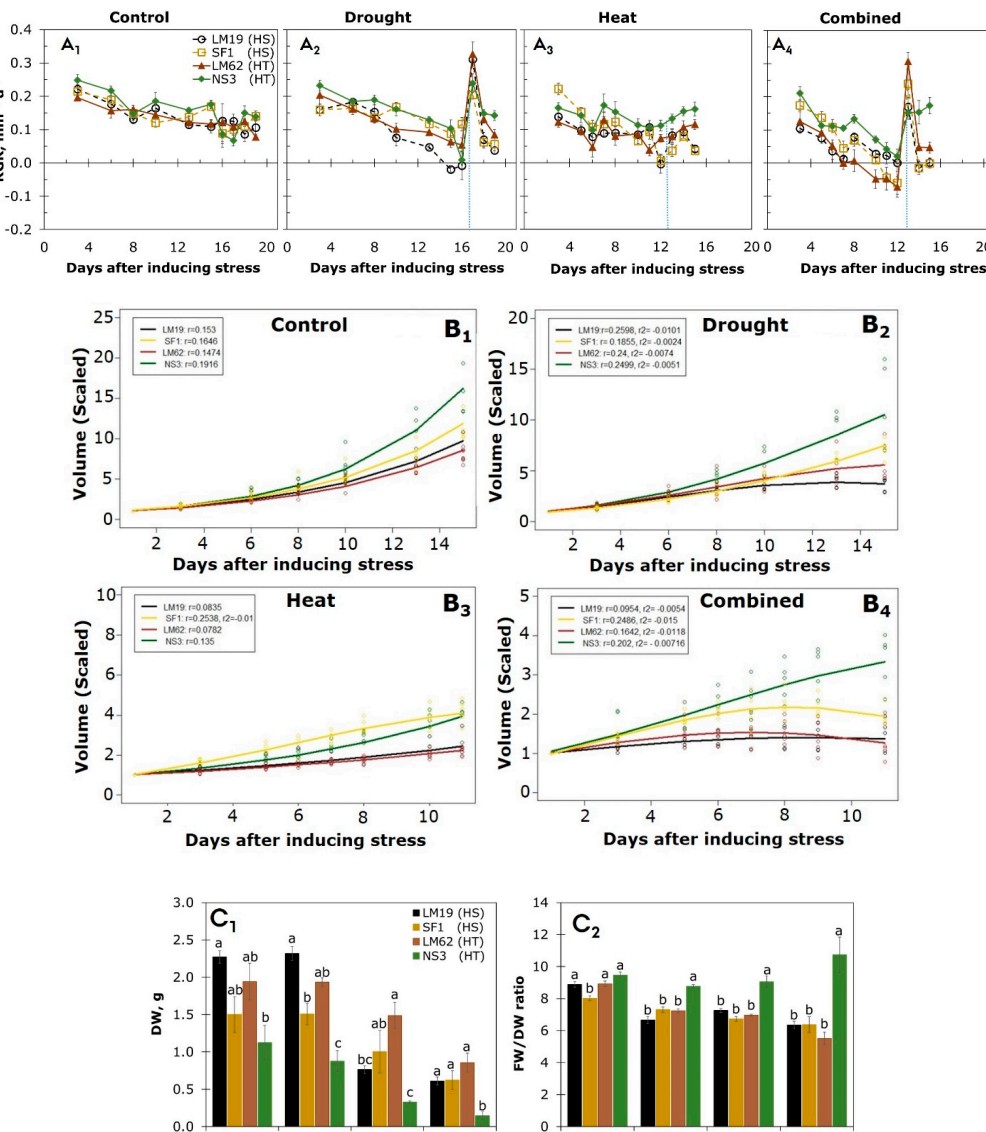

**Figure 3.** (**A$_{1–4}$**) Relative growth rate (RGR) along days after inducing the stress treatments followed by three days of recovery in four wheat genotypes. HS represents heat-sensitive genotypes (open symbols, dashed lines) and HT represents heat-tolerant genotypes (closed symbols, solid lines). Blue dotted line denotes the recovery phase. The data represent mean values ± S.E.M. (*n* = 5). (**B$_{1–4}$**) Growth models for plant volume in different treatments. Bold lines represent the models and circles point out the scaled data. In the legends the estimated coefficients of the growth models are presented (Equations (1) and (2) described in materials and methods). (**C$_{1,2}$**) Dry weight (DW) and ratio of fresh weight/dry weight (FW/DW) from destructive harvest at the last day after inducing the stress. The data represent mean values ± S.E.M. (*n* = 4). Different small letters indicate significant differences in each genotype by using Tukey's test at level *p* < 0.05.

The growth modeling was applied based on the calculated plant volume to describe the specific growth performance strategies of the genotypes in given conditions (Figure 3B$_{1–4}$ and Table S3.2). The values of the coefficients resulting from the models, combined with the visualization of the curves and pairwise comparisons test using mixed models for the scaled volume, clearly showed that NS3 was the most resistant genotype to the single and combined stress, despite the fact that NS3 was the smallest genotype in size. However,

the growth curve in NS3 was always above the other genotypes. In the control treatment, this means that it grew faster (higher $r$ coefficient). In the stress treatments, a bell-shaped curve was the best fit for the combination of growth ($r$) and decay ($r_2$) (Figure 3B legends). With heat stress, SF1 was modeled differently from the others, as it showed the best growth in the early phase of stress treatment (Figure 3B$_3$), but in the late phase clearly displayed the decay of the curve ($r_2$ coefficient is significant as the bell-shaped model was a better fit). In addition, the pairwise comparisons tests using mixed models for the scaled volume showed that both SF1 and NS3 were significantly higher than LM19 and LM62 at the late stress phase (Table S3.2). The same was observed under combined stress, and SF1 showed the highest reduction among all the genotypes used (Figure 3B$_4$, $r_2$ value of SF1 was more negative).

Dry weight (DW) was measured at the last day of stress by destructive harvest (Figure 3C$_1$). DW was similar in all genotypes in the control treatment. With drought stress, the difference was observed within both the heat-sensitive and -tolerant genotypes. DW was significantly higher in LM19 compared to SF1, and was higher in LM62 compared to NS3. With heat and combined stress, this difference was found within heat-tolerant genotypes only. DW was significantly higher in LM62 compared to NS3. Moreover, the DW was significantly lower only in NS3 compared to the other genotypes, similar to the leaf area calculated from RGB imaging data under combined stress (Figures 3C$_1$ and S4A–C$_{1-4}$ and Table S7). In the control treatment, the fresh and dry weight ratio (FW/DW) was significantly lower in SF1, while in all stress treatments the FW/DW was significantly higher in NS3 compared to the other genotypes (Figure 3C$_2$). A strong linear correlation was observed between both the fresh and dry weights and the image-based data at the last day of stress in all genotypes (Figure S5A$_{1-6}$,S5B$_{1-6}$). In addition, the water consumption and plant volume were correlated for the respective treatments (Figure S6).

### 3.3. Chlorophyll Fluorescence Imaging

Chlorophyll fluorescence imaging was conducted to assess the physiological status of the plants under individual and combined stress (Figure 4 and Table S4). In the control treatment, the PSII operating efficiency ($F_q'/F_m'$) initially had a higher level in the heat-tolerant LM62 and NS3 compared to the heat-sensitive LM19 and SF1 genotypes, but later, throughout the experiment, no difference was observed between the genotypes (Figure 4A$_1$ and Table S4.1). In all genotypes, $F_q'/F_m'$ decreased at the last day of drought stress and was significantly lower only in the heat-sensitive genotype LM19 compared to the other genotypes, but increased again in the recovery phase (Figure 4A$_2$). The $F_q'/F_m'$ was significantly lower during late heat stress and the recovery in the heat-sensitive compared to the heat-tolerant genotypes (Figure 4A$_3$). With combined stress, a pronounced difference was observed in $F_q'/F_m'$ during the late stress phase between genotypes. During the late combined stress and recovery phases, NS3 maintained a significantly high $F_q'/F_m'$, while a reduction was observed in the other genotypes (Figure 4A$_4$). The $F_q'/F_m'$ decreased gradually in LM62 and SF1 compared to LM19 and NS3. In spite of the fact that the $F_q'/F_m'$ was higher in LM19 after day 9 compared to LM62 and SF1, during the recovery phase, $F_q'/F_m'$ recovered better in LM62 than in SF1 (Figure 4A$_4$). In all genotypes, the fraction of open PSII centers in steady state ($q_L$) increased in the experimental period of the control and drought stress (Figure 4B$_{1,2}$ and Table S4.2). With heat stress, $q_L$ was maintained along the days after inducing stress, and it was higher in NS3 compared to the other genotypes (Figure 4B$_3$). However, no difference was observed between the genotypes as regards recovery from heat stress (Figure 4B$_3$). With combined stress, $q_L$ decreased after day 8 in LM62 followed by SF1 and NS3, while in LM19, the $q_L$ decreased after day 12. NS3 showed the highest $q_L$ in combined stress and the recovery phase (Figure 4B$_4$).

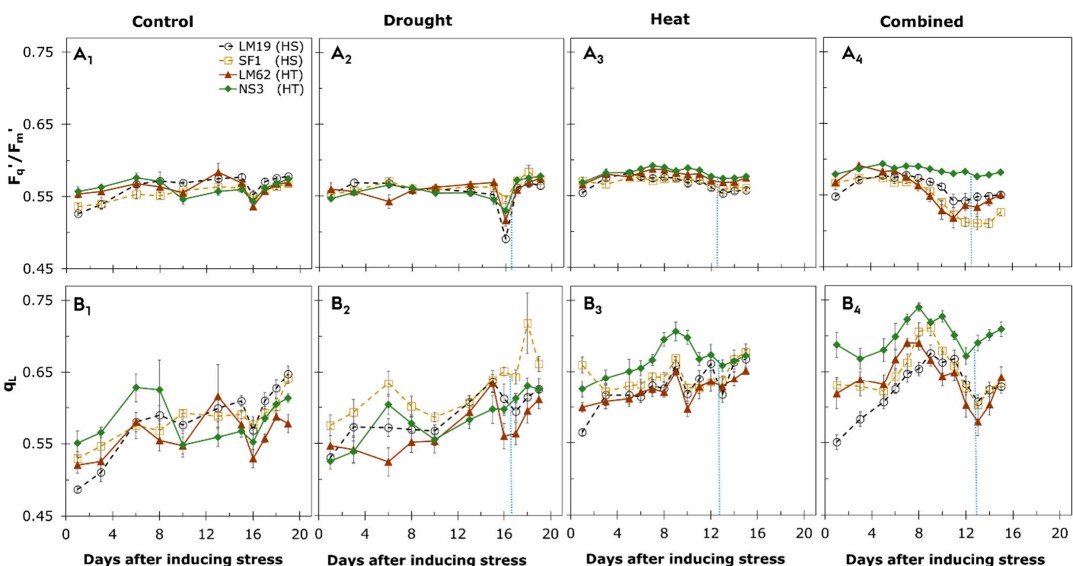

**Figure 4.** Chlorophyll fluorescence measurements on light-adapted plants at PPFD of 750 µmol m$^{-2}$ s$^{-1}$. (**A$_{1-4}$**) PSII operating efficiency ($F_q'/F_m'$). (**B$_{1-4}$**) Fraction of open PSII centers (oxidized Q$_A$) in steady state (q$_L$) at day 1 before inducing the stress and along days after inducing the stress treatments in four wheat genotypes. HS represents heat-sensitive genotypes and HT represents heat-tolerant genotypes. Blue dotted line denotes the recovery phase. The data represent mean values ± S.E.M (*n* = 5).

By dark-adapting the plants for 25 min followed by a light response curve, a difference was observed within the treatments and genotypes compared to the non-dark-adapted plants. The maximum quantum efficiency of PSII photochemistry in dark-adapted plants ($F_v/F_m$) was maintained in the control and drought treatments (Figure 5A$_{1,2}$ and Table S5.1). A similar response in terms of $F_v/F_m$ of the dark-adapted leaves (using the FluorPen FP100, PSI, Drásov, Czech Republic) was observed where there were no significant differences in control and drought treatments in all genotypes (Figure 5E). With heat stress, a pronounced difference was observed in $F_v/F_m$, with significantly lower ratios in NS3 compared to LM62 (Figure 5A$_3$). With combined stress, the $F_v/F_m$ was significantly lower in NS3 only during stress, but not at the last day of stress (Figure 5A$_4$). By comparing with the $F_v/F_m$ measured by FluorPen FP100 at the last day of heat stress, the $F_v/F_m$ was higher in the heat-tolerant compared to the heat-sensitive genotypes. During combined stress, the $F_v/F_m$ was significantly higher in the heat-tolerant NS3 compared to LM62 (Figure 5E).

From the light response curves (Figure S7 and Table S8) at a PPFD of 750 µmol m$^{-2}$ s$^{-1}$, the $F_q'/F_m'$ was maintained under control and drought treatments in all genotypes except at day 8, and the last day of drought stress, $F_q'/F_m'$ was significantly lower in NS3 compared to the other genotypes (Figure 5B$_{1,2}$ and Table S5.2). As a consequence of the heat and combined stress treatments, the $F_q'/F_m'$ increased immediately at day 3 after inducing the stress, and decreased slightly at the last day of combined stress in all genotypes (Figure 5B$_{3,4}$). However, the heat-tolerant genotype NS3 showed higher $F_q'/F_m'$ compared to the heat-sensitive genotypes LM19 and SF1 at the last day of heat stress (Figure 5B$_3$). The q$_L$ and non-photochemical quenching (NPQ) were maintained as well in the control and drought stress treatments in all genotypes, except in NS3, where the q$_L$ was lower compared to the other genotypes at the last day of drought stress (Figure 5C$_{1,2}$,D$_{1,2}$ and Tables S5.3 and S5.4). In all genotypes, the q$_L$ was maintained during heat and combined stress, while q$_L$ decreased slightly at the last day of combined stress (Figure 5C$_{3,4}$). NPQ decreased immediately at day 3 after inducing heat and combined stress, and then it remained stable in all genotypes in the beginning and the end of the stress treatments (Figure 5D$_{3,4}$).

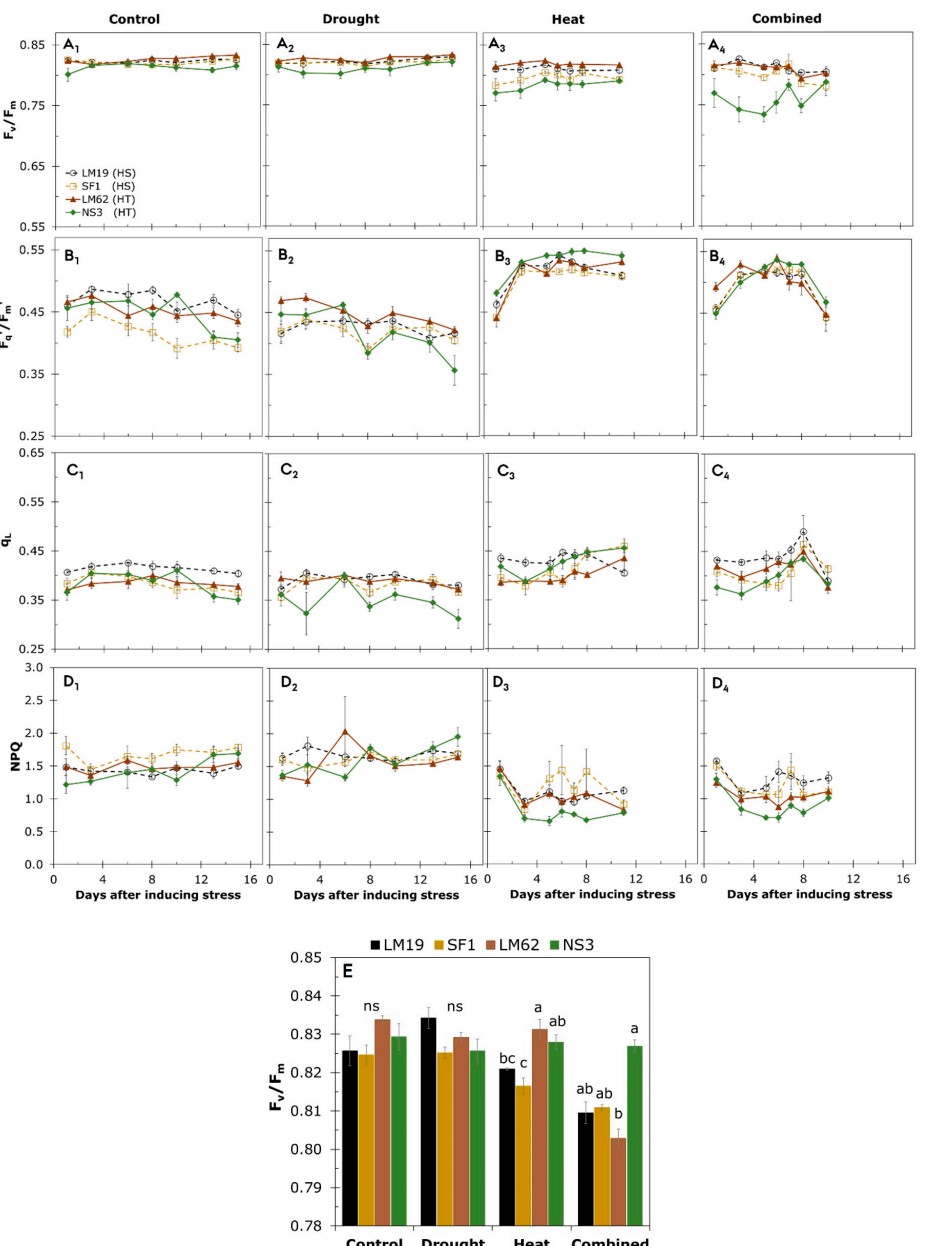

**Figure 5.** Chlorophyll fluorescence measurements on dark-adapted plants. (**A$_{1-4}$**) Maximum quantum efficiency of PSII photochemistry ($F_v/F_m$). From light response curves at PPFD of 750 µmol m$^{-2}$ s$^{-1}$. (**B$_{1-4}$**) PSII operating efficiency ($F_q'/F_m'$). (**C$_{1-4}$**) Fraction of open PSII centers (oxidized $Q_A$) in steady state ($q_L$). (**D$_{1-4}$**) Non-photochemical quenching (NPQ) at day 1 before inducing the stress and along days after inducing the stress treatments in four wheat genotypes. HS represents heat-sensitive genotypes and HT represents heat-tolerant genotypes. The data represent mean values ± S.E.M. (*n* = 4). (**E**) $F_v/F_m$ measured manually on dark-adapted leaves with Fluor-Pen FP100 at the last day of stress treatments in four wheat genotypes. The data represent mean values ± S.E.M. (*n* = 4). Different small letters indicate significant differences in each genotype by using Tukey's test at level *p* < 0.05.

### 3.4. Thermal IR Imaging and Water Use Efficiency

The leaf cooling (ΔT) measurement is directly dependent on leaf transpiration, which reflects the stomatal regulation when measured under the same vapor pressure deficit irrespective of treatment. In the control treatment, ΔT reduced gradually in all genotypes, indicating increased leaf cooling (Figure 6A$_1$). During drought stress, ΔT increased after day 10 and decreased after recovery except in SF1, which maintained cooler leaves (Figure 6A$_2$

and Table S6.1). In all genotypes, the $\Delta T$ decreased gradually after inducing heat stress and increased slightly after recovery (Figure 6A$_3$). With combined stress, a reduction in $\Delta T$ was observed in the early phase (first week) of stress in all genotypes. $\Delta T$ increased in LM62 after day 5, followed by SF1, while LM19 was increased $\Delta T$ after 8 days of inducing combined stress (Figure 6A$_4$). After recovery, the $\Delta T$ showed a decrease in all genotypes (Figure 6A$_4$).

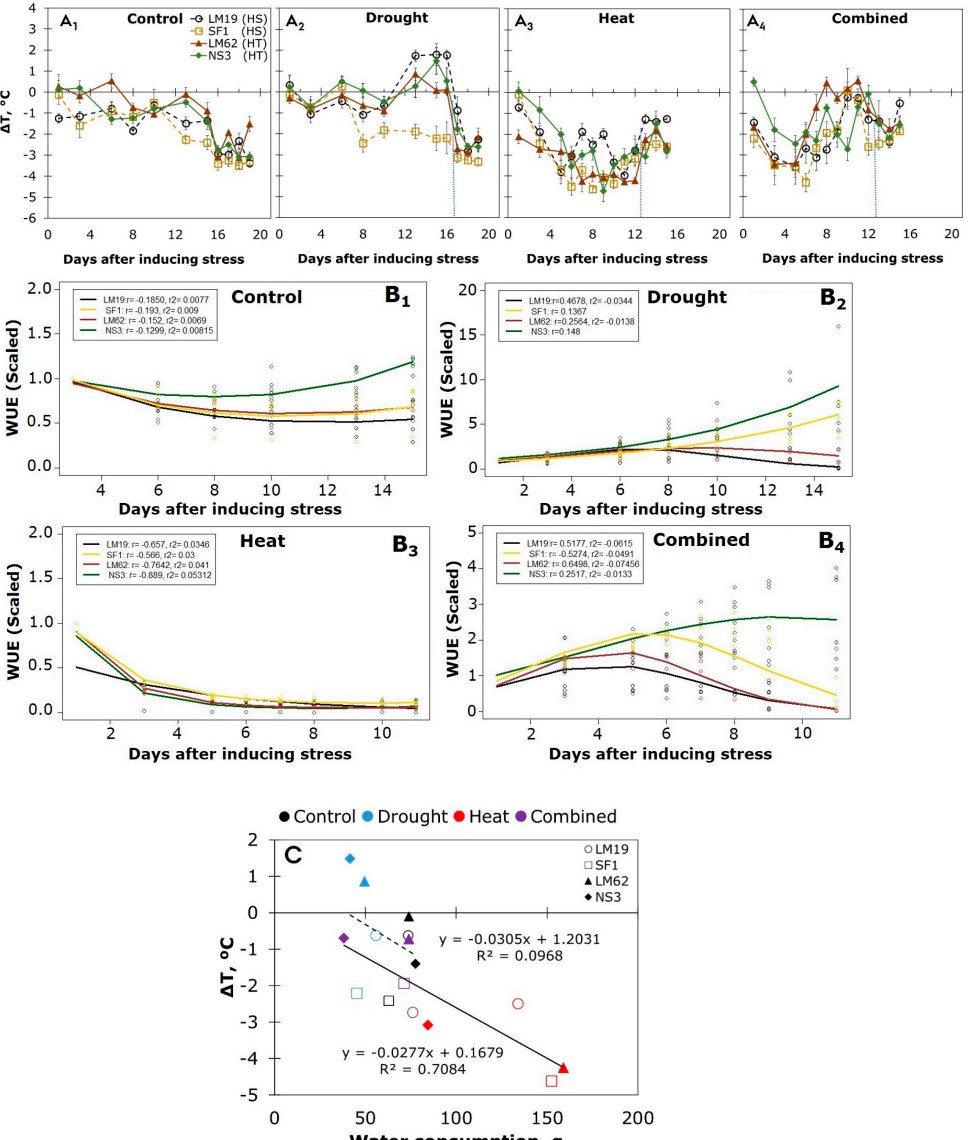

**Figure 6.** Canopy temperature. (**A$_{1-4}$**) Difference between canopy average and air temperature measured in the thermal IR imaging unit ($\Delta T$) at day 1 before inducing the stress and along days after inducing the stress treatments, followed by three days of recovery in four wheat genotypes. HS represents heat-sensitive genotypes (open symbols, dashed lines) and HT represents heat-tolerant genotypes (closed symbols, solid lines). Blue dotted line denotes the recovery phase. The data represent mean values $\pm$ S.E.M. ($n = 5$). (**B$_{1-4}$**) Growth models for water use efficiency (*WUE*) in different treatments. Bold lines represent the models and the circles point out the scaled data. In the legends, the estimated coefficients of the growth models are presented (Equations (1) and (2) described in materials and methods). (**C**) Correlation between $\Delta T$ and water consumption at the last day of four different treatments in each wheat genotype. The linear dashed line represents control and drought treatments and the linear solid line represents heat and combined stress treatments. $R^2$ values represent the level of fit for the linear regressions ($n = 4$).

The water use efficiency (*WUE*) of the plants was assessed throughout the experiment and the growth models for the different genotypes were calculated in the given conditions. The growth models for *WUE* reflected the cumulative water amount used every day per plant biomass produced (Figure 6B$_{1-4}$). As expected, the best *WUE* for all genotypes was under drought and combined stress treatments (Figure S6A$_{2,4}$). The growth modeling with *WUE* scaled values showed that NS3 and SF1 in the early phase of stress maintained a better *WUE* under drought and combined stress treatments (Figure 6B$_{2,4}$). However, SF1 showed better *WUE* only in the early phase of combined stress. Moreover, LM19 and LM62 had lower *WUE*, similar to the volume growth modeling results (Figure 3B$_{2,4}$), at the end of the stress period under drought and combined stress treatments, compared to SF1 and NS3. During the heat stress treatment, though, a similar *WUE* was observed at the late stage of heat stress for all genotypes. Genotype pairwise comparisons for *WUE* based on mixed models confirmed that NS3 had higher *WUE* at the later stage of combined stress compared to the other genotypes (Table S6.2). Moreover, the water consumption and ΔT under heat and combined stress were correlated, showing that the plants with lower ΔT consumed more water as observed in LM62 and SF1 with heat stress. With combined stress, the heat-tolerant genotypes had similar ΔT values, with different water consumption and higher ΔT compared to the heat-sensitive genotypes (Figure 6C). Additionally, by analyzing the data through correlation heatmap, strong correlations can be shown to exist among the measured morpho-physiological traits of the same category (Figure S3).

### 3.5. Leaf Pigment Analysis

The measurement of spectral reflectance from leaves was used to investigate the vegetation indices of different stress treatments. The photochemical reflectance index (PRI) was significantly lower in LM62 compared to NS3, regardless of the treatments (Figure 7A$_1$). A pronounced difference was observed between the heat-tolerant genotypes, as the normalized difference vegetation index (NDVI) was significantly lower in LM62 in the control and drought stress, compared to NS3 (Figure 7A$_2$). For further investigation, leaf samples were harvested for analysis of changes in plant pigment contents under the different treatments (Figure 7B). During drought stress, some of the leaf pigment contents were significantly higher in SF1 and NS3 compared to the other genotypes. SF1 had significantly higher violaxanthin, neoxanthin and lutein contents, and violaxanthin + antheraxanthin + zeaxanthin pigments (V + A + Z), while NS3 had significantly higher zeaxanthin and β-carotene, and both showed higher chlorophyll a+b contents under drought stress (Figure 7B$_{1-8}$). The heat-tolerant genotype subjected to heat stress showed higher zeaxanthin, lutein, β-carotene and chlorophyll a+b contents compared to the heat-sensitive genotypes (Figure 7B$_{3,5-7}$). With combined stress, the heat-tolerant genotypes showed higher violaxanthin and chlorophyll a+b contents and V + A + Z compared to the heat-sensitive genotypes (Figure 7B$_{1,7,8}$). Even though the leaf pigment contents in SF1 were kept high under drought and heat stress, a reduction was observed in all pigment contents with a significant increase in both chlorophyll a/b ratio and degree of de-epoxidation of xanthophyll-cycle components ((Z + A)/(V + A + Z)) under combined stress (Figure S8A$_{1,2}$). A strong correlation was observed between β-carotene and chlorophyll a+b contents (Figure S8B). In LM19 and LM62 with drought and in SF1 with combined stress treatments, the pigment contents were lower compared to the other genotypes. With heat and combined stress, heat-tolerant genotypes were higher in comparison to the heat-sensitive genotypes (Figure S8B).

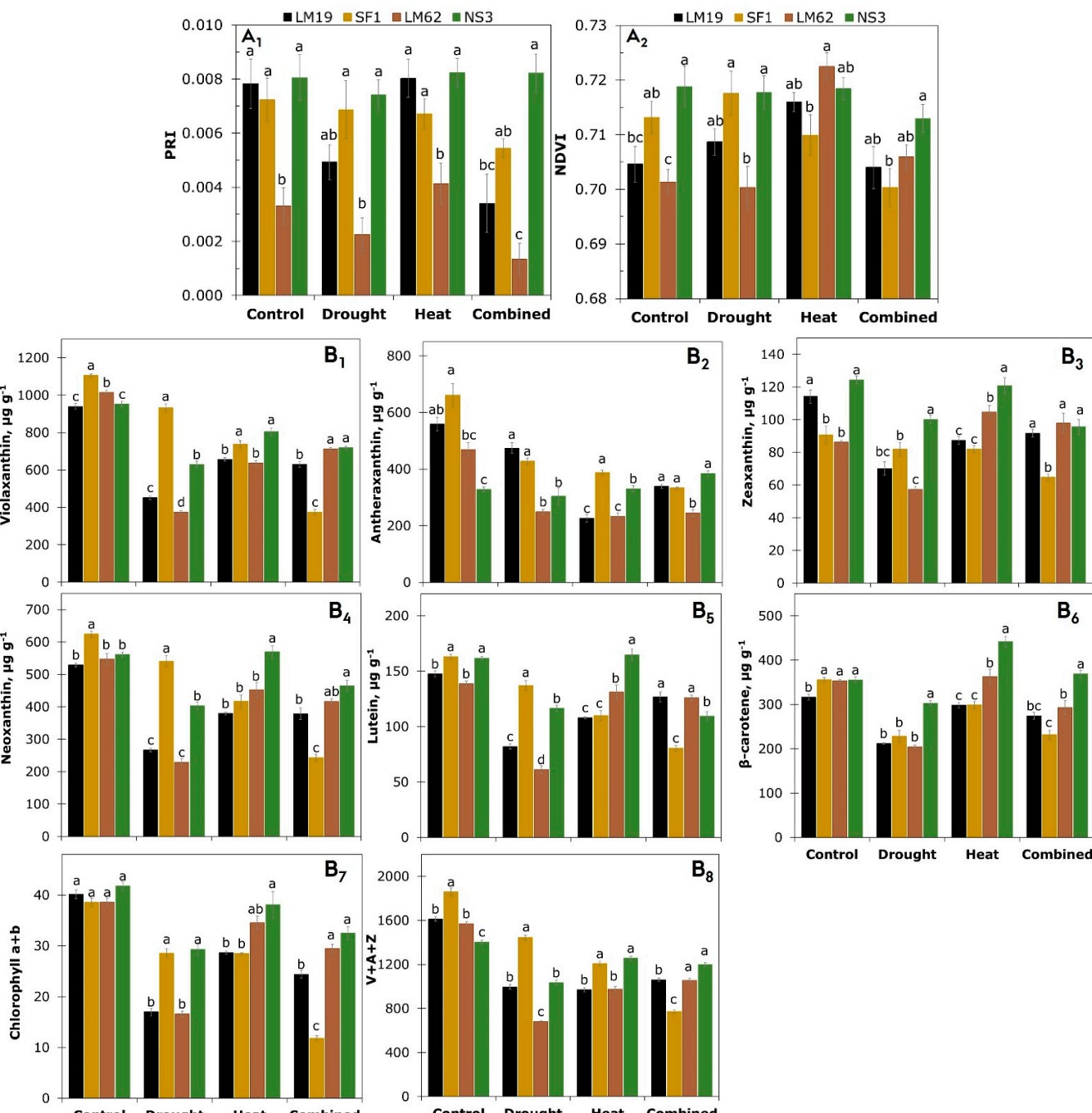

**Figure 7.** Leaf pigment measurements. (**A₁,₂**) Photochemical reflectance index (PRI) and normalized difference vegetation index (NDVI) determined with Polypen RP410 at the last day of four different treatments in four wheat genotypes. The data represent mean values ± S.E.M. ($n = 9$). (**B₁–₈**) Leaf pigment contents at the last day of stress treatments in four wheat genotypes. The data represent mean values ± S.E.M. ($n = 4$). Different small letters indicate significant differences in each genotype by using Tukey's test at level $p < 0.05$.

## 4. Discussion

By assessing the digital biomass and plant growth rates through the RGB imaging and applying exponential growth models, differences between the heat-sensitive and -tolerant genotypes were observed, regardless of the applied stress treatments under controlled conditions. LM19 and LM62 were in principle exposed to more severe drought stress, due to their higher biomass, the larger transpiring leaf area and lower SRWC compared to SF1 and NS3. Thus, at the late drought stress phase the RGR was lower in LM19 and LM62,

whereby a negative RGR in the latter reflected a reduction in leaf area with increasing drought severity [12]. This is similar to a previous study with wheat cultivars under well-watered and drought stress conditions, whereby different water consumption and physiological responses were observed, indicating different adjustments of stomatal closure, and variation in resource allocation strategies and growth development [60]. Heat-tolerant wheat genotypes had higher stomatal conductance and transpiration rates to lower the leaf temperature, thereby mitigating the negative effect of heat stress, which was accompanied by higher dry matter accumulation [61]. Our findings correspond to this as the RGR was significantly higher at the last day of heat stress in the heat-tolerant genotypes LM62 and NS3, and increased in the recovery phase compared to the heat-sensitive genotypes LM19 and SF1. However, a different growth performance was observed in the heat-tolerant genotypes in the heat and combined stress treatments, whereby DW was significantly higher in LM62 compared to NS3, suggesting that LM62 rapidly consumed water to maintain a cooler leaf surface while NS3 preserved water. This was further confirmed by the growth models for *WUE*, whereby NS3 had better *WUE* in the control and combined stress treatments. This might be due to the fact that NS3 had a faster growth rate.

Moreover, NS3 showed the smallest biomass and highest FW/DW ratio in all stress treatments, and maintained high RGR under individual and combined stress, which is one of the reasons for the lack of response to severe stress compared to the other genotypes. With combined stress, the SRWC decreased drastically in all genotypes, except in NS3, which had the smallest biomass of all genotypes. The exponential growth models showed that NS3 was the most resistant to both individual and combined stress, despite the fact that NS3 was the smallest genotype in size. At the last day of combined stress, the RGR was lower in LM62 and SF1 compared to NS3, reflecting different mechanisms when plants are exposed to the combined stress compared to the individual stress [25]. Overall, under combined stress the growth performance of SF1 was strongly affected in the late phase of the stress, as shown by the exponential growth models of plant volume and *WUE*, reflecting that SF1 was more susceptible to combined stress. In this study, the RGB imaging results were strongly correlated with plant biomass harvested at the last day of stress, corresponding to biomass harvested in tomato [48].

The photochemical efficiency of photosystem II and photosynthetic performance were investigated by the chlorophyll fluorescence measurements [35]. In this study, PSII operating efficiency in light-adapted plants ($F_q'/F_m'$) was unaffected in the early drought stress phase and decreased only during the late drought stress phase (after 10 days of stress). $F_q'/F_m'$ was significantly lower in LM19 compared to the other genotypes, showing that LM19 was more susceptible to drought, as SRWC was lower with higher DW during the late drought stress phase. At the late moderate heat stress and recovery phase, $F_q'/F_m'$ was significantly lower in heat-sensitive genotypes compared to the heat-tolerant genotypes, corresponding to previous results where the latter maintained higher maximum quantum efficiency of PSII ($F_v/F_m$) under heat stress compared to the heat-sensitive genotypes [62]. During early combined stress, $F_q'/F_m'$ was not affected as the drought effect was not pronounced, similar to barley, wherein photosynthetic performance was affected under heat and combined stress, but not under drought stress, as the genotypes had different mechanisms to adapt to combined stress conditions compared to the individual stress [63]. By increasing the stress severity, a reduction in the electron transport rate occurred and decreased the ability of the photosynthetic apparatus to maintain oxidized $Q_A$ [35]. We found that $F_q'/F_m'$ decreased in the late combined stress phase, where the drought effect was slightly pronounced except in NS3. However, the heat-sensitive genotype LM19 showed higher $F_q'/F_m'$ compared to LM62 and SF1. The heat-tolerant LM62 recovered better than the heat-sensitive SF1, suggesting that the damage was not severe and reversible. Moreover, the fraction of open PSII centers ($q_L$) decreased in LM62, followed by SF1 and NS3, while in LM19 the $q_L$ decreased after day 12 even if all genotypes recovered, showing that LM19 had an improved response under the combined stress.

A difference in the effect of the stress treatments was observed by using different protocols for light-adapted and dark-adapted plants. $F_v/F_m$ measured by fluorescence imaging was unaffected by the stress treatments regardless of the tolerance level. $F_v/F_m$ showed no significant difference in control and drought treatments in all genotypes, which is probably due to the mild stress effect. It has been reported that $F_v/F_m$ is not affected under mild drought stress as a result of increased photorespiration, which maintains the electron transport rate and protects PSII from damage when stomatal limitation occurs [36]. It is noteworthy that the variation found between the values of $F_v/F_m$ measured by imaging and those measured by FluorPen on young fully expanded leaves was due to the fact that the imaging covers the $F_v/F_m$ on the entire plant. Since $F_v/F_m$ is sensitive to light stress, the light gradient will affect the average of $F_v/F_m$ measured on the entire plant, as it was reported previously that a slight gradient of $F_v/F_m$ was observed along the leaf length [62]. When measurements of $F_v/F_m$ were taken on an exposed young fully expanded leaf, the light level was similar for all measured plants regardless of the canopy size. Thus, the $F_v/F_m$ from imaging was lower in NS3, as they were smaller plants with less internal shading, and the lower $F_v/F_m$ in NS3 was probably related to the small plant size and not to the stress treatment.

The NPQ has an important role in protecting the photosynthetic apparatus from photoinhibition by dissipating excess excitation energy as heat [64]. To analyze the NPQ, light response curves were measured after dark-adaptation. In this study, the induced stress was mild. Thus, the $F_q'/F_m'$ and $q_L$ from the light response curves were maintained regardless of the treatments, and decreased slightly at the last day of combined stress in all genotypes. Similarly, a reduction in $F_q'/F_m'$ and $q_L$ occurred under combined stress in tomatoes regardless of the heat tolerance potential, indicating that heat-tolerant cultivars might be more susceptible under the combined stress [29]. The photosynthetic apparatus is protected under drought stress through the occurrence of alternative electron sinks, and with increasing drought severity the NPQ increased in order to avoid photoinhibition [65]. However, NPQ was maintained in the control and drought treatments in all genotypes, while NPQ decreased slightly in all genotypes at day 3 after inducing the heat and combined stress, but it was maintained at the end of the stress phase.

The stay-green trait is a sign of stress tolerance, and expanding the stay-green duration under stress has a positive effect on maintaining the grain-filling period [61]. Heat-tolerant wheat varieties have better leaf-cooling capacity, as shown by the more pronounced depression of canopy temperature [66]. The canopy temperature is a vital physiological trait that could be used in screening heat-sensitive and -tolerant genotypes, as it reflects both $CO_2$ assimilation and stomatal regulation of the canopy [67]. Plant canopy temperature was investigated here by thermal IR imaging. In drought-stressed plants, $\Delta T$ increased after day 10 in all genotypes except in SF1, showing that SF1 might have different stomatal regulation. By contrast, the $\Delta T$ decreased in all genotypes during heat stress due to the high transpiration rate, indicating enhanced leaf cooling at high temperatures [15,68]. A difference in the canopy temperature depression was also observed in soybean genotypes under control and drought stress, showing that differences in canopy development between genotypes affect the transpiration rate [69]. Similarly, in our study the response differed between the heat-sensitive and -tolerant genotypes, whereby SF1 and LM62 consumed more water and had lower $\Delta T$ compared to the other genotypes, probably due to the difference in biomass between the genotypes. The increased $\Delta T$ under combined stress in the heat-tolerant genotype LM62 after day 6 indicated that LM62 was more susceptible to combined stress, as expressed by the reduction of RGR, $F_q'/F_m'$ and PRI.

The heat stability of PSII and photosynthetic apparatus has been interpreted as the acclimation of the wheat plants to the higher temperature [70]. In agreement with this, a better recovery was observed in the heat-tolerant LM62, despite being more susceptible in the combined stress (as observed by reductions in RGR, $F_q'/F_m'$ and $q_L$ and increases in $\Delta T$), compared to the heat-sensitive genotypes, suggesting that the decrease in these parameters was reversible and PSII was not damaged. In addition, the correlation between

the water consumption and ΔT showed that ΔT was higher in heat-tolerant genotypes under drought and combined stress, indicating a better stomatal regulation to avoid excess water loss.

Leaf pigment analysis was conducted at the last day of stress to investigate possible photoprotection mechanisms, i.e., the xanthophyll cycle and the conversion from violaxanthin to zeaxanthin [71]. A decrease in chlorophyll a and carotenoid pigments under drought stress was reported in a drought-sensitive wheat cultivar [72]. In accordance, the two genotypes SF1 and NS3 were less susceptible to drought stress and had higher contents in most pigments compared to LM19 and LM62. One of the adaptive responses of plants exposed to mild heat stress is adjusting the carotenoids to maintain the photosynthetic apparatus as found in potato [73]. The heat-tolerant genotypes showed, for most pigments, higher contents under heat and combined stress compared to the heat-sensitive genotypes. A previous study found in the drought-sensitive wheat genotype an increase in $(Z + A)/(V + A + Z)$ under drought stress, indicating an increase in the energy dissipation, while non-significant changes were observed in the composition of $V + A + Z$ pigments in the drought-tolerant genotype [53]. Our results are in accordance with this, whereby a reduction in $V + A + Z$ and chlorophyll a+b contents was found with drought stress in heat-sensitive LM19 compared to SF1 and in heat-tolerant LM62 compared to NS3, and with combined stress in SF1, which was accompanied by a reduction in the measured morpho-physiological traits. Only SF1 showed significantly high $V + A + Z$ pigment levels, reflecting better tolerance to individual drought and heat stress, but not with the late phase of combined stress. With heat and combined stress, chlorophyll a+b contents increased in the heat-tolerant genotypes, reflecting the adaptation of PSII to maintain the photosynthetic apparatus [54]. Moreover, the reduction in pigment contents in LM19 and LM62 under drought and in SF1 under combined stress corresponded with the RGB and chlorophyll fluorescence parameters due to the stress susceptibility. The photochemical reflectance index (PRI) is dependent on the developmental stage and canopy structure, and is strongly correlated with the green leaf area index and chlorophyll contents under moderate stress [74]. The PRI was significantly lower under drought and combined stress in LM62, reflecting that they were exposed to the stress before the other genotypes by reduction in the SRWC and RGR at the late stress phase.

## 5. Conclusions

The selected Nordic wheat genotypes grown under controlled conditions showed variation in their tolerance to individual and combined drought and heat stress. LM19 and LM62 were susceptible to a later drought stress episode, as indicated by the exponential growth models, lower RGR and $F_q'/F_m'$ and higher ΔT, as well as changes in both the contents and compositions of pigments. The moderate heat stress, particularly the early stress episode, did not result in significant differences between the heat-tolerant and -sensitive genotypes. A pronounced difference was observed with the late heat stress event, where the heat-tolerant genotypes showed higher RGR and $F_q'/F_m'$, and higher contents of some pigments such as zeaxanthin and β-carotene, compared to the heat-sensitive genotypes. With the combined stress, a higher RGR in the later phase of stress treatment contributed to the partly improved growth performance in the heat-sensitive genotype LM19. By contrast, SF1 had improved tolerance to individual drought or heat stress episodes as indicated by lower ΔT and high RGR, but it was more susceptible to combined stress primarily in the late stress phase. NS3 was the only genotype that showed higher RGR and FW/DW ratios, and maintained higher $F_q'/F_m'$ and $q_L$ with heat and combined stress, because of their relatively small biomass compared to the other genotypes. This correlated with the higher *WUE* under combined as well as drought stress. As such, the heat-tolerant and -sensitive genotypes reflected different susceptibilities to the individual and combined stress. This study provides insights into the dynamic changes and physiological responses of different wheat genotypes to the interactive effects of individual

and combined abiotic stress types. In addition, the applied phenotyping approach was found to be useful for selecting robust crops for a future, more variable climate.

**Supplementary Materials:** The following are available online at https://www.mdpi.com/2073-439 5/11/2/364/s1, Table S1: Measured phenotypic traits in this study by using PlantScreen™ Modular system (PSI, Drásov, Czech Republic). Tables S2–S8: Pairwise comparisons test using mixed models to determine the significant differences between the means of four wheat genotypes in different time points for all figures. Figure S1: Scheme of thermal infrared (IR) imaging processing. Figure S2: Classification of the measured phenotypic traits into categories. Figure S3: Trait correlation via heatmap analysis. Figure S4: Analyzed side and top view RGB imaging. Figure S5: Correlation between fresh and dry weight with RGB data. Figure S6: Water use efficiency and correlation between plant volume and water consumption at the last day of stress. Figure S7: Light response curves after dark adaptation. Figure S8: Leaf pigment analysis.

**Author Contributions:** Conceptualization, L.O.A.A., E.R., B.W., C.-O.O. and K.P.; methodology, investigation and formal analysis, L.O.A.A. and K.P.; resources, K.P.; data curation, L.O.A.A., I.S. and K.P.; software, I.S.; visualization, L.O.A.A., E.R., B.W., C.-O.O. and K.P.; writing—original draft preparation, L.O.A.A.; writing—review and editing, L.O.A.A., E.R., B.W., I.S., C.-O.O. and K.P.; supervision, E.R., B.W., C.-O.O. and K.P.; project administration, E.R. and C.-O.O.; funding acquisition, I.S., E.R. and C.-O.O. All authors have read and agreed to the published version of the manuscript.

**Funding:** This project was in part funded by a project from the Levy foundation, Promillaafgiftsfonden, The research school for Science and Technology, and iClimate (Interdisciplinary Center for Climate Change) at Aarhus University. In addition, it was supported by European Regional Development Fund-Project "SINGING PLANT" (No. CZ.02.1.01/0.0/0.0/16_026/0008446) with financial contribution from the Ministry of Education, Youths and Sports of the Czech Republic through the National Programme for Sustainability II funds.

**Institutional Review Board Statement:** Not applicable.

**Informed Consent Statement:** Not applicable.

**Data Availability Statement:** Data is contained within the article and Supplementary Materials.

**Acknowledgments:** We thank Pavla Homolová for helping with the preparation of plant material and Jaromír Pytela for technical support during the experiments at Photon Systems Instruments (PSI) Research Center (Drásov, Czech Republic). We thank Radka Kočí for running the HPLC analysis.

**Conflicts of Interest:** The authors declare no conflict of interest.

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
