# Peer review of "Investigating Combined Drought- and Heat Stress Effects in Wheat under Controlled Conditions by Dynamic Image-Based Phenotyping"

_agronomy, doi:10.3390/agronomy11020364_

Round 1

Reviewer 1 Report

The manuscript "Investigating the combined drought- and heat stress effect on 2 physiological traits in Nordic wheat genotypes by dynamic im-3 age-based phenotyping" by Abdelhakim et al. is very interesting. With a set of new phenotyping methods the authors analyzed a set of four cultivars, which are expected to be different in their behavior to heat/drought stress. Four lines with only a few plants is not enough to draw important conclusion about general differences of stress tolerance. However, the authors showed how the methods could be used to get information and showed the way to analyze more plants.

But I have also a concern regarding the statistical analysis of the data. As the authors have a tremendous amount of data, which can be organized by time points, treatments (even heat and drought stress and their interaction) and cultivars, there was not attempt  to analyze the data statistically with classical methods like ANOVA or mixed models. I would at least expected that in the supplementary material the readership can find these analyses. There are so many figures, which show developments, but it has not been presented, whether the development  in time was statistically significant. Furthermore it would be very important to know whether the found effects of heat and drought are different from the control, as the control did also show movements in trait values over time. So, as there are several statistical methods to analyze the effects and the development of effects I would like to recommend that the authors add these analyses and discuss the data in the light of the extra information.

Overall, I think that the study should be ok when this concern is addressed.  

Author Response

We appreciate the effort that you have dedicated for your insightful comments on the manuscript. Please see the attachment.

Reviewer 2 Report

Lamis et al. presented a study using imaging and physiological measurements to uncover responses of different wheat varieties to heat/drought stress. The experimental design was rigorous and conclusions were well supported. All referred methods were well introduced. I only have a few suggestions to improve the manuscript.

  1. Please use day and DAS consistently in the manuscript for readers to better track;
  2. For subpanels in figure 2b, you need scale bars to indicate the size of pixel;
  3. Figure 3b1, you missed “15” on y-axis;
  4. Figure 6c, it is better to indicate the size of dot in the figure but not the legend;
  5. L492, should be “leaf pigment analysis”

Author Response

(The authors gave the same response as above.)
